# Proportion of active tuberculosis among HIV-infected children after antiretroviral therapy in Ethiopia: A systematic review and meta-analysis

**Fassikaw Kebede Bizuneh**[1]*, **Dejen Tsegaye**[1], **Belete Negese Gemeda**[2], **Tsehay Kebede Bizuneh**[3]

**1** College of Health Sciences, Debre Markos University, Debre Markos, Ethiopia, **2** College of Health Sciences, Debre Berhan University, Debre Berhan, Ethiopia, **3** College of Social Science Bahir Dare University, Bahir Dare, Ethiopia

* fassikaw123@gmail.com, fishmar216@gmail.com

**Data Availability Statement:** All relevant data are within the paper and its Supporting Information files.

## Abstract

Despite effectiveness of antiretroviral therapy in reducing mortality of opportunistic infections among HIV infected children, however tuberculosis (TB) remains a significant cause for morbidity and attributed for one in every three deaths. HIV-infected children face disproportionate death risk during co-infection of TB due to their young age and miniatures immunity makes them more vulnerable. In Ethiopia, there is lack of aggregated data TB and HIV mortality in HIV infected children. We conducted an extensive systematic review of literature using Preferred Reporting of Systematic Review and Meta-Analysis (PRISMA) guideline. Five electronic databases were used mainly Scopus, PubMed, Medline, Web of Science, and Google scholar for articles searching. The pooled proportion of TB was estimated using a weighted inverse variance random-effects meta-regression using STATA version-17. Heterogeneity of the articles was evaluated using Cochran's Q test and I2 statistic. Subgroup analysis, sensitivity test, and Egger's regression were conducted for publication bias. This met-analysis is registered in Prospero-CRD42024502038. In the final met-analysis report, 13 out of 1221 articles were included and presented. During screening of 6668 HIV-infected children for active TB occurrence, 834 cases were reported after ART was initiated. The pooled proportion of active TB among HIV infected children was found 12.07% (95% CI: 10.71–13.41). In subgroup analysis, the Oromia region had 15.6% (95%CI: 10.2–20.6) TB burden, followed by southern Ethiopia 12.8% (95%CI: 10.03–15.67). During meta-regression, missed isoniazid Preventive therapy (IPT) (OR: 2.28), missed contrimoxazole preventive therapy (OR: 4.26), WHO stage III&IV (OR: 2.27), and level of Hgb $\leq$ 10gm/dl (OR = 3.11.7) were predictors for active TB. The systematic review found a higher proportion of active TB in HIV-infected children in Ethiopia compared to estimated rates in end TB strategy. To prevent premature death during co-infection, implement effective TB screening and cases tracing strategies in each follow up is needed.

**Funding:** The authors received no specific funding for this work.

**Competing interests:** The authors have declared that no competing interests exist.

**Abbreviations:** TB, uberculosis; WHO, World Health Organization; FMOH, Federal Ministry of Health; HIV, human immune deficiency virus; HAAR, highly active antiretroviral therapy; IPT, isoniazid preventive Therapy; CPT, co-trimoxazole preventive therapy.

# Introduction

People living with the Human Immune deficiency virus (PLHIV) are more susceptible to tuberculosis (TB), which is a leading cause of mortality [1, 2]. There is a strong synergy between HIV infection and TB, while PLHIV is at high risk of dying from TB and HIV infection is the biggest risk factor for active TB incidence through declining cellular immunity and increased endogenous reactivation of latent TB bacilli in the lungs [3, 4]. HIV infected children are at increased risk of acquiring active TB. HIV-infected persons are sixteen times more likely to be co-infected by TB disease as compared to HIV-negative person [5].

Tuberculosis continued to be the leading cause of morbidity and mortality for people living with HIV (PLHIV) worldwide [6]. Globally, in 2022, an estimated 1.3 million children (aged 0–14 years) were diagnosed with TB, accounting for approximately 12% of the total TB cases of 10.6 million [5, 7]. The co-infection of HIV and TB is particularly dangerous, with around 214,000 children dying from TB disease in 2022 where 31,000 of those were attributed to children TB and HIV infections [5]. The burden of TB infection varies significantly across each continents, African and Southeast Asian regions attributed for 81% of global TB deaths in 2022 [8]. In Sub-Saharan African countries, 10% to 15% of HIV-infected children suffer from the dual burden of HIV and TB, with a lifetime risk of 21% and two-thirds of cases remain undiagnosed [5, 9–11].

By the end of 2022, only 46% of children (aged 0–14 years) who were receiving antiretroviral therapy (ART) were able to achieve viral load suppression, which is a crucial factor in reducing the occurrence of new opportunistic infections [5, 8, 11]. However, TB infection remains responsible for one in every third deaths of HIV infected children in source-limited setting [7, 12, 13]. Ethiopia is one of the top 30 countries burdened by tuberculosis (TB) and experiences a significant distribution of TB and HIV co-infection across all regions. The incidence rates was estimated as 0.17 cases per 1000 population for HIV and 1.64 per 1000 for TB [1, 2]. Previous national level study finding among 1,830,880 HIV and 192,359 TB patients reported,7.34% of TB patient had HIV infection with a significant regional variation across regions [14]. The prevalence of TB/HIV co-infection varies considerably in across each regions including 7.2% in Amhara region (Northern Ethiopia) [15] to 23.6% southern Ethiopian (SNNR) [16]. The differences in healthcare accessibility and socio-demographic factors including wealth index and literacy rate contribute to variations in TB/HIV co-infection prevalence [14]. HIV-infected children face a higher risk of morbidity and mortality during co-infection due to their young age and immature immune makes them more vulnerable [4, 17].

Previous studies finding in Ethiopia [7, 18, 19] reported that multifactorial causative factors were attributed for active TB occurrence among HIV infected children including underweight, advanced WHO clinical stages, missed IPT and CPT [5, 7, 13, 20]. However, CD4 count being ≤200 cells/ml serves as a proxy indicator for incidence of active TB [13]. Concomitant administration of ART with isoniazid preventive Therapy (IPT) had significantly effect of reducing active TB cases by over 80% HIV infected Children [21]. However, IPT completion rate and adherence of ART has affected by caregivers and regimen characteristic [18]. Although several small-scale studies have reported on the epidemiology of TB/HIV co-infection among HIV-infected children in various parts of Ethiopia [5, 12, 13]; however, there is a lack of aggregated data on co-infection after HIV-infected children started antiretroviral therapy. Therefore, this systematic review and meta-analysis aimed to estimate the pooled burden of active TB among HIV-infected children.

# Methods

## Study area and setting

This study was conducted in Ethiopia from January 1, 2013, to December 30, 2022, spanning a period of 10 years. In Ethiopia, there are nine regions including Tigray, Afar,

Amhara, Oromia, SNNR, Somalia, Gambella, Benishangul Gumuz, Harari and two city administrative [13].

**Searching strategy and protocol.** The Preferred Reporting Items for Systematic Reviews and Meta-Analysis (PRISMA) guideline was followed to report the findings of the selected articles presented clearly described in (**S1 Checklist**) [22].

Additionally, this systematic review and meta-analysis have registered in the Prospero protocol with CRD42024502038 (https://www.crd.york.ac.uk/prospero/#recordDetails).

Furthermore, this systematic review was used five international electronic databases were mainly used including Scopus, PubMed, Medline, Web of Science, and Google scholar. The searching was focused on English language published articles and the searching was done. We employed controlled vocabulary terms (MeSH) and free text to extract articles (**S1 Text**).

The search included topics such as active tuberculosis, pulmonary TB, extra pulmonary TB, HIV infection, individuals, children, pediatrics, neonates, lymphadenitis, disseminated TB, and Ethiopia. The search terms used to identify relevant studies included "Epidemiology" OR "Incidence" OR "Case fatality" "Tuberculosis" OR "Pulmonary Tuberculosis" OR "Disseminated Tuberculosis" OR "Lymphadenitis" AND "HIV" OR "AIDS" AND "Children" OR "Pediatrics" OR "Infant" AND "Ethiopia". Furthermore, this systematic review and meta-analysis employed the PICO (Population, Intervention, Comparison, and Outcomes) framework to assess the eligibility of the articles and enhance evidence-based medicine and research by facilitating the structuring of clinical or research questions. This included as follows (P) Population of interest: Children living with HIV on anti-retroviral therapy in Ethiopia,(I) Intervention; all children HIV infected children started Anti-retroviral therapy, (C) Comparison; children without active TB with stand on HIV cohort (O) Outcome of interest: active TB in HIV-infected children found in Ethiopia were used for PICO frameworks.

## Eligibility criteria

**Inclusion criteria.** This systematic review and meta-analysis report had included a given articles with defined outcome of any TB types in HIV infected children with the following inclusion criteria. 1) scientific papers reporting co-infections of TB and HIV in HIV-infected children in Ethiopia, 2) articles containing burden or incidence reports of active TB in HIV-infected children, 3) studies published within the past ten years with cross-sectional or cohort designs **and published** in English, and 4) study subjects limited to children aged ≤15 years.

**Exclusion criteria.** Studies that reported lacking abstracts and/or full-text, anonymous reports, editorials, and qualitative studies were excluded from the analysis. Furthermore, prior to the analysis, unfitted articles without a journal name and/or author, lacked the year of publication, and citations without abstracts and/or full-text were removed.

**Outcome ascertainment.** The first outcome was the proportion of active TB cases (including all types of TB) among HIV infected children after anti-retroviral therapy. The proportions of TB burden was calculated by the number of children who developed active TB during on ART treatment divided by the total children from thirteen study and multiplied it by 100. Identifying independent predictors for active TB occurrence in HIV infected children on ART was the second objective. Accordingly, we collected significant predictors reported from included articles with their adjusted odd ratio with its 95% confidence interval was extracted from original studies and to computed the pooled odds ratio for final predictors.

## Operational words

**Advanced HIV disease.** Defined as WHO clinical stages III and IV in children older than five years. However, in children younger than five years living with HIV, they are considered

to have advanced HIV disease regardless of their clinical stages. Mild WHO clinical stages refer to stages II and I in HIV-positive children. **ART adherence** for children; is categorized as follows: Good (>95%) if ≤2 doses are missed out of 30 doses or ≤3 doses out of 60 doses and Fair (85–94%) if 3–4 doses are missed out of 30 doses and poor (<85%) if >5 doses are missed out of 30 doses of ART drug [23].

**Data extraction.**  Four Authors (FK, BN, DT, and TK) extracted articles and evaluated the quality of each study by determining the eligibility based on given criteria for selection of studies. The discussion was used to settle any disagreement or uncertainty that arose during the article extraction and removing duplication process. These reviewers assessed the full-text articles; if one or more of them believed an article could be significant, it qualified after the article was carefully examined its titles, abstracts, and full text by three authors (FK, TK, and DT) used a Microsoft Excel spreadsheet to extract the specifics of each article. Three independent reviewers assessed each included article's quality using the JBI checklists given for all articles as described in (**S1 Table**) [24, 25]. All eligible studies approved by all authors' agreements about principal investigators, year of publication, study period, study setting, study population, and sample size retrieved from the identified articles. The biases of primary studies checked, assessed and screened by three authors (FK, BN and TK), evaluated, and screened (S2 Table). Any disagreements among reviewers regarding the critical appraisal were settled through discussion and building consensus for submission.

**Software and statistical-analysis.**  Using End-Note Aversion 8.1, all detected and potentially suitable published article citations were exported and gathered; duplications were eliminated during the selection and screening processes. Two independent reviewers (FK, and TK) first reviewed the abstracts of the publications before moving on to the full-text articles, which they then evaluated following the particular standards for ultimate inclusion and modifying the data on a Microsoft Excel spreadsheet, and employed the STATA version 17 for further analysis. Descriptive statistics, and weighted inverse variance random-effects meta-regression were used to present the review's results to estimate pooled burden of active TB in HIV infected children [26]. The eligible articles were extracted using Meta-XL Excel version 5.3sheet [27] using identified risk factors from each selected studies and made combined each categorical variables and estimated risk factors for active TB [26]. The Higgs $I^2$ statistics were utilized to detect heterogeneity among studies and elaborated using Cochran's Q test [28]. The degrees of statistical heterogeneity between the studies were assessed using I2 statistics; values of 25%, 50%, and 75% were thought to indicate modest, medium, and high levels of heterogeneity, respectively [38]. The source of heterogeneity among studies was examined using the subgroup and sensitivity analysis. The random effect regression model was used for the data-identified heterogeneous analysis [26]. The publication biases were assessed by visual inspection of funnel plots of the graph and quantitative using Egger's weighted regression at p <0.1 [29, 30].

## Results

### Descriptive characteristics of the studies

A total of 1221 primary studies were identified including 43 from Web of Science, 631 from PubMed, 352 from Medline, 15 from Scopus, and 162 articles from Google Scholar. After care full screening throughout the articles titles and abstracts, 1208 articles excluded. Thirteen (N = 13) individual studies that met inclusion criteria were included for the final meta-analysis reported [4, 15–17, 31–39] as presented and described in PRISMA diagram (**Fig 1**).

Regarding to include articles description all are published in scientific journals from December 30, 2012 to January 1st, 2023. Regionally seven(N = 7)of articles among eligible

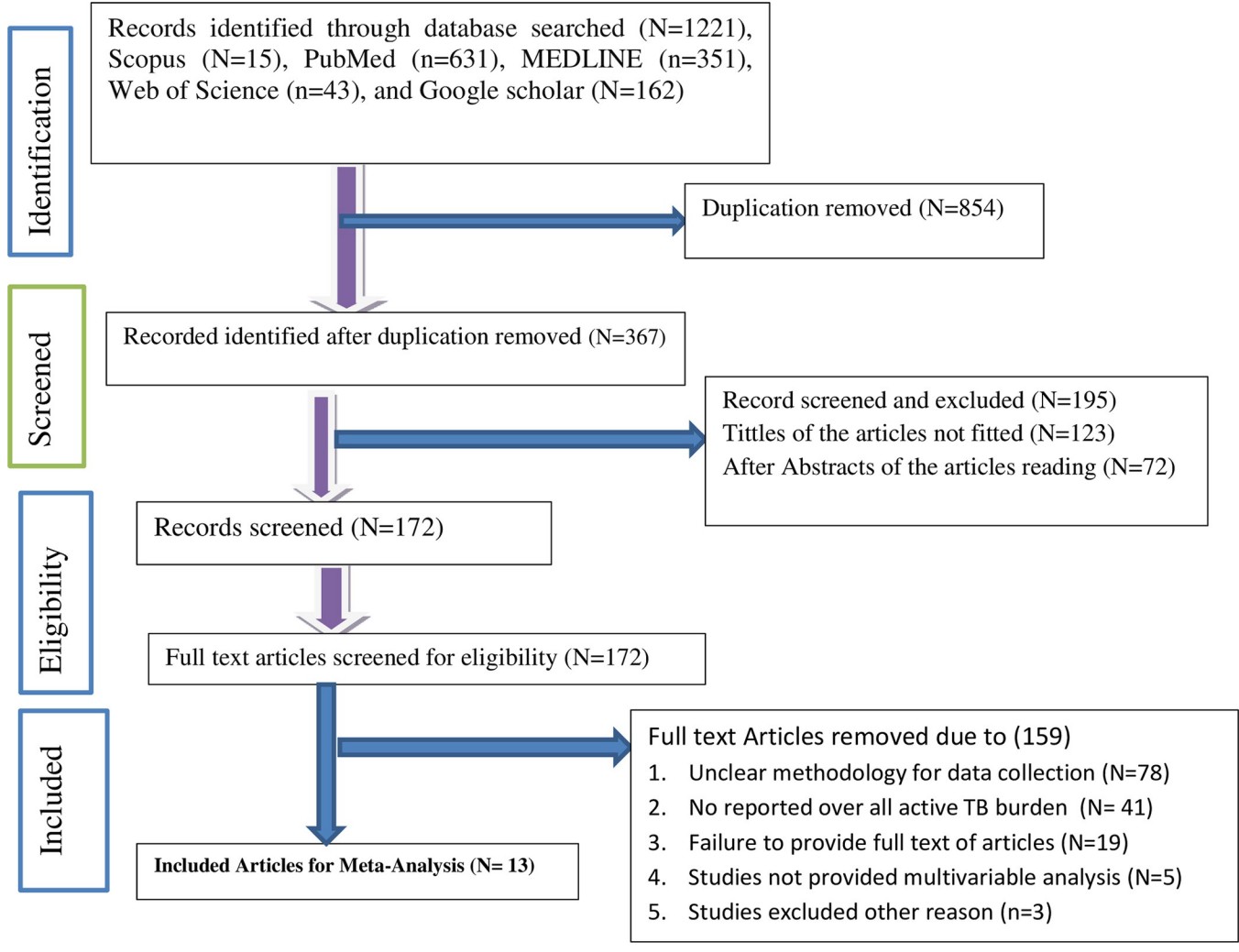

**Fig 1. PRISMA flow diagram for articles searching diagram.**

articles were from Amhara region (Northern parts of Ethiopia) [15, 31–33, 38, 39] and three articles were from southern nation nationalities region (SNNRs) of Ethiopia [16, 34], two of the remaining articles were from Benishangul Gumuz (North west) part of Ethiopia [4, 17], and one were from Oromia [37] regions which is clearly described in (**Table 1**).

## Description reports of included studies

From included 13 individual studies, 6668 HIV infected children were participated and 834 cases of TB among HIV, infected children reported. The mean (±SD age of the participants was reneged from 8.2(±3.6) years. Of the total, nine(9/13) included articles were employed cohort design [16, 17, 32, 34, 38, 39], whereas four of the included articles data were collected by correctional recorded review of follow up design [4, 15, 31, 35, 38] were used. The highest number of active TB cases (23.6%) was reported from the SNNPR region (Southern Ethiopia) [16] and the lowest number (7.2%) of active TB cases was from Amhara region (North West Ethiopia) [15] regions respectively.

**Table 1. Characteristics of included articles/studies reporting the prevalence of TB in HIV-positive children in Ethiopia after ART initiation in Ethiopia.**

| Full first author name | Year | Region | Design | Sample size | MA/Years | Events (TB cases) | Proportion (%) | Incidence/100 PPY | Follow up time | Study setting | Quality |
|---|---|---|---|---|---|---|---|---|---|---|---|
| Yihun Mulugeta Alemu et al. [31] | 2016 | Amhara | Follow up | 647 | 6 | 79 | 12.2 | 4.2 | 5 | HT&HC | 3 |
| Fassikaw Kebede Bizuneh et al. [17] | 2021 | Benishangul | Cohort | 421 | 8 | 52 | 12.4 | 5.9 | 5 | HT | 3 |
| Firew Tiruneh et al. [16] | 2020 | SNNRS | cohort | 800 | 9 | 189 | 23.6 | 7.9 | 5 | HT&HC | 3 |
| Fassikaw Kebede Bizuneh et al. [4] | 2021 | Benishangul | Cohort | 428 | 10.3 | 64 | 14.9 | 5.78 | 10 | HT&HC | 3 |
| Firew Tiruneh et al. [35] | 2020 | SNNr | Cohort | 844 | 9 | 113 | 13.4 | 3.36 | 5 | HT&HC | 3 |
| Aklilu Endalamaw et al. [30] | 2018 | Amhara | Cohort | 352 | 6.7 | 34 | 9.6 | 2.63 | 13 | HT | 3 |
| Beshir Masino Tessu et al. [36] | 2019 | Oromia | Cohort | 428 | 6 | 67 | 15.6 | 6.03 | 5 | HT | 3 |
| Sualiha Gebeyaw Ayalaw et al. [32] | 2015 | Amhara | Follow up | 271 | 9.8 | 44 | 16.2 | 4.9 | 6 | HT | 3 |
| Mamaru Wubale Melkamu et al. [37] | 2020 | Amhara | Cohort | 408 | 6.3 | 42 | 10.3 | Not Reported | 15 | HT&HC | 3 |
| Ermias Sisay Chanie et al. [15] | 2022 | Amhara | Follow up | 349 | 7.3 | 25 | 7.2 | Not Reported | 11 | HT | 3 |
| Dagnaw Amare Mequanente et al. [38] | 2022 | Amhara | Cohort | 389 | 7.9 | 57 | 10.5 | Not Reported | 6 | HT | 3 |
| Endalk Birrie Wondifraw et al. [19] | 2022 | Amhara | Follow up | 358 | 8.3 | 57 | 15.9 | 2.0 | 14 | HT | 3 |
| Emil Westerlund et al. [33] | 2014 | Arba Minch | Cohort | 139 | 5.9 | 11 | 7.9 | Not Reported | 6 | HT | 2 |

\* HC = Health center, HT = hospital, PPY = person per years, **Mean age/Y =** Years, SNNR = south nation, nationalities' region

## Pooled prevalence of TB in HIV infected children on ART

In the final meta-analysis report, utilizing 13 published studies, we discovered that the estimated pooled burden of active TB among HIV-infected children in Ethiopia was 12.1% (95% CI: 10.7–13.4; $I^2$ = 63.4%, P = 0.001) as described in (**Fig 2**).

## Factors associated with active TB in HIV infected children on ART

In our final report, there was significant heterogeneity observed among the studies included in the meta-analysis (I2 = 63.4%, P <0.001 as depicted in pooled proportion of active TB in HIV infected children. Accordingly, the pooled TB prevalence was slightly lower in hospital setups at 11.05% (95%CI: 9.4–12.3) compared to health center studies, which reported 14.1% (95%CI: 11.74–16.33) (**Fig 3**). Likewise, the pooled TB burden among HIV-infected children was significantly higher in studies conducted in the Oromia region at 15.6% (95%CI: 10.2–20.6) compared to studies included from the SNNR, which had a result of 12.8% (95%CI: 10.03–15.67) as described in (**Fig 3**).

In this report, the duration of follow-up periods was found to be significantly associated with the occurrence of active TB. Sub-group analysis revealed that the pooled burden of TB among HIV-infected children with a follow-up period of ≤10 years was significantly higher at 13.67% (95%CI: 11.24–15.1) compared to those with a follow-up period of >10 years, which had estimation of 10.9% (95%CI: 9.1–12.8) as described in (**Fig 4**).

In this systematic review, to identify factors associated with active TB we analyzed adjusted odds ratios from primary studies and made grouped significant categorical variables from

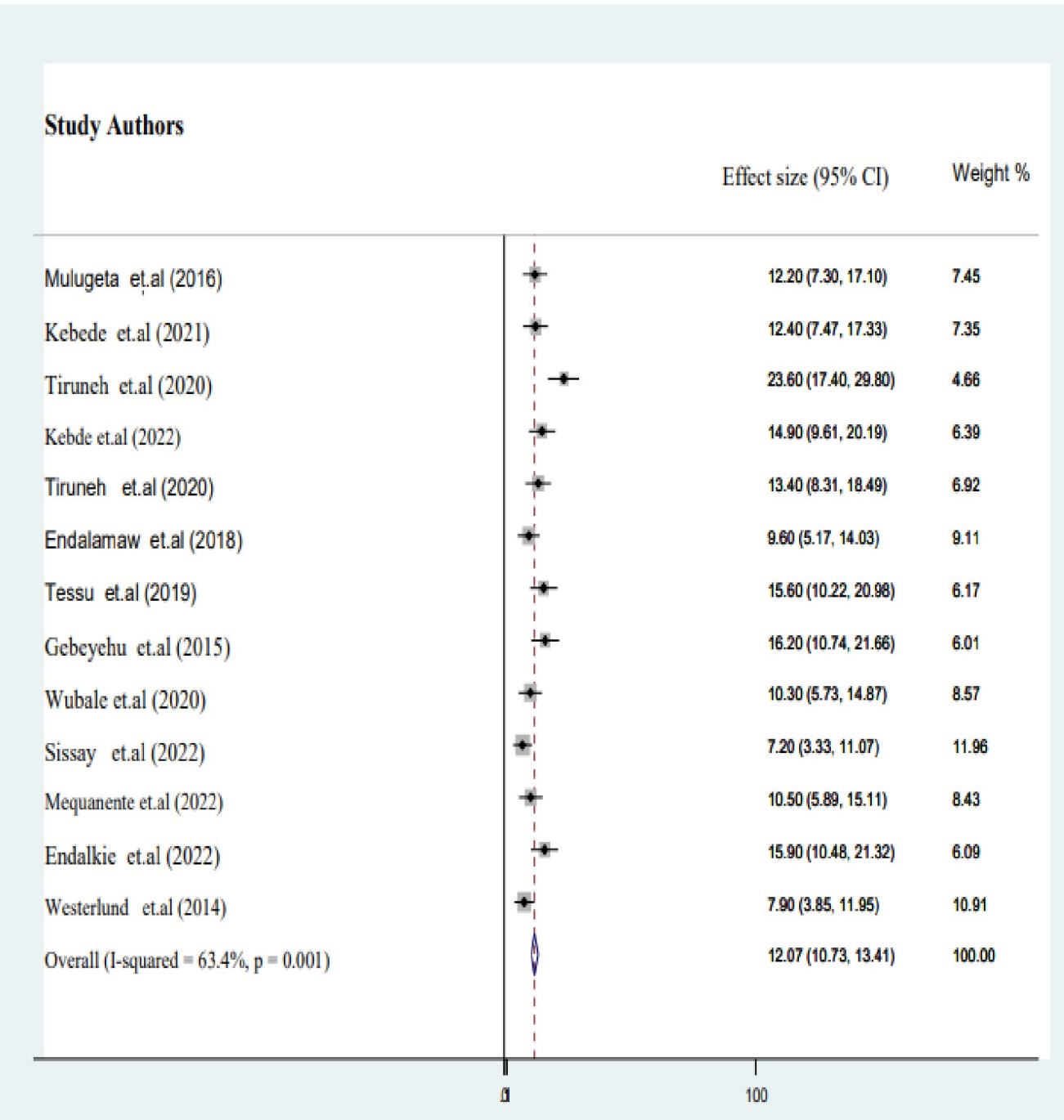

**Fig 2. Forest plot of pooled proportion of active TB among HIV-infected children in Ethiopia.**

previous studies by themes, including WHO advanced clinical stages (III&IV), baseline CD4 count, missed isoniazid preventive therapy (IPT), missed cotrimoxazole preventive therapy (CPT), level of hemoglobin, antiretroviral therapy (ART) adherence status, and functional status of children. But, it is noted that only missed IPT, missed CPT, WHO advanced clinical stages (III&IV) and level of hemoglobin were found predictors for TB as shown in (**Table 2**).

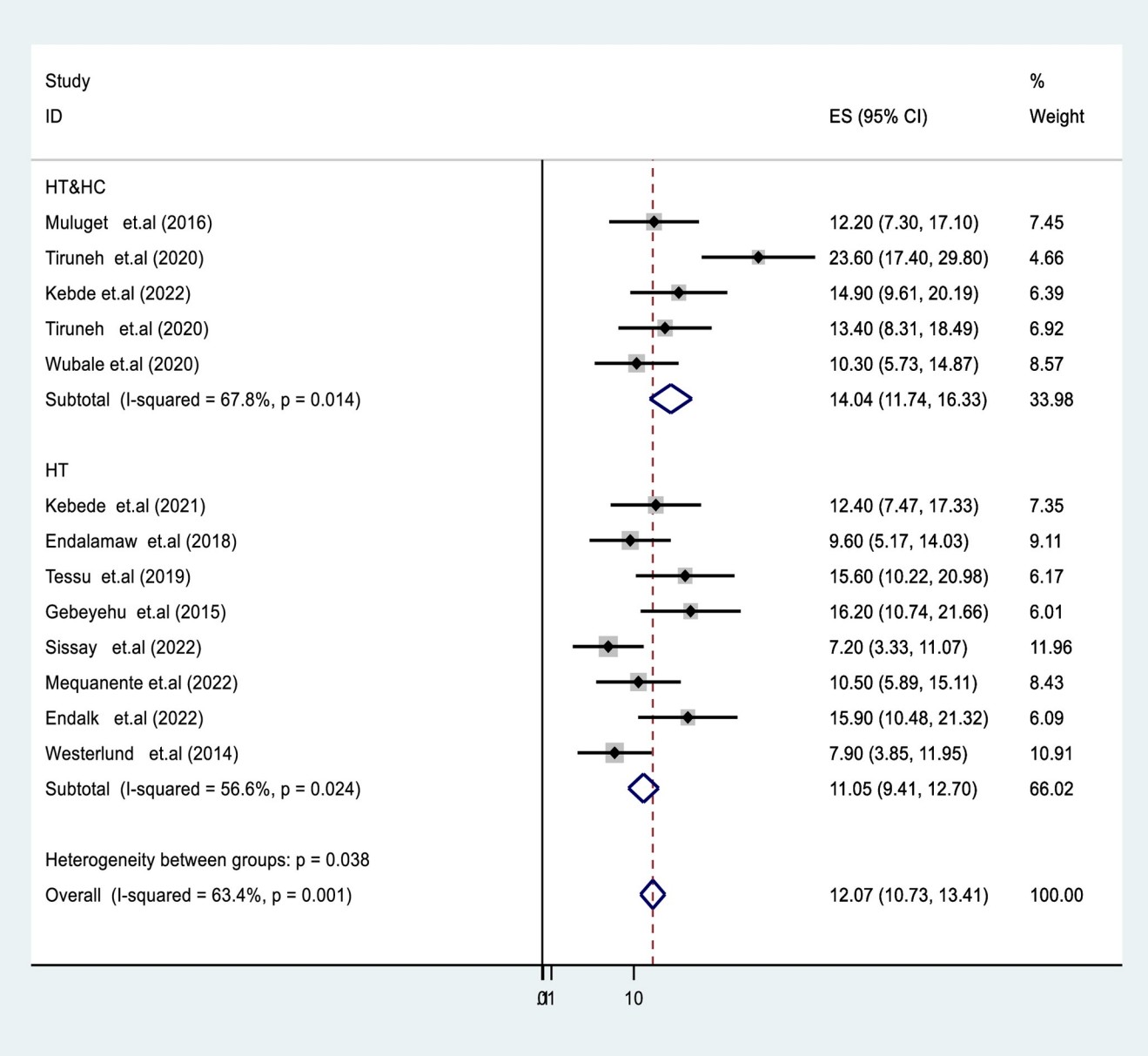

*HC= health center; HT=Hospital

**Fig 3. Forest plot for subgroup analysis by study setting of active TB proportion in HIV-infected children in Ethiopia.**

Accordingly, studies containing HIV infected children who missed IPT was double fold increase the odds of active TB occurrence compared with ever given children (OR: 2.28; 95% CI: 1.99–3.08) and also the likelihood of active TB occurrence in HIV infected children who are on advanced WHO clinical stage (III&IV) was 2.27 times (OR: 2.27; 95% CI: 1.81–2.73) higher than with children were on WHO clinical stage II and I. Furthermore, the probability of TB co-infection for HIV-infected children was 3.11 times higher (OR = 3.11, 95% CI: 1.57–4.7) for cases having hemoglobin≤10 mg/dl compared to children with a Hgb >10 mg/dl as described in (**Fig 5**). Furthermore HIV-infected children who missed CPT had 4 time higher odds of TB co-infection than counter group (OR: 4.26, 95% CI: 3.47–5.28) clearly depicted in (**Fig 6**).

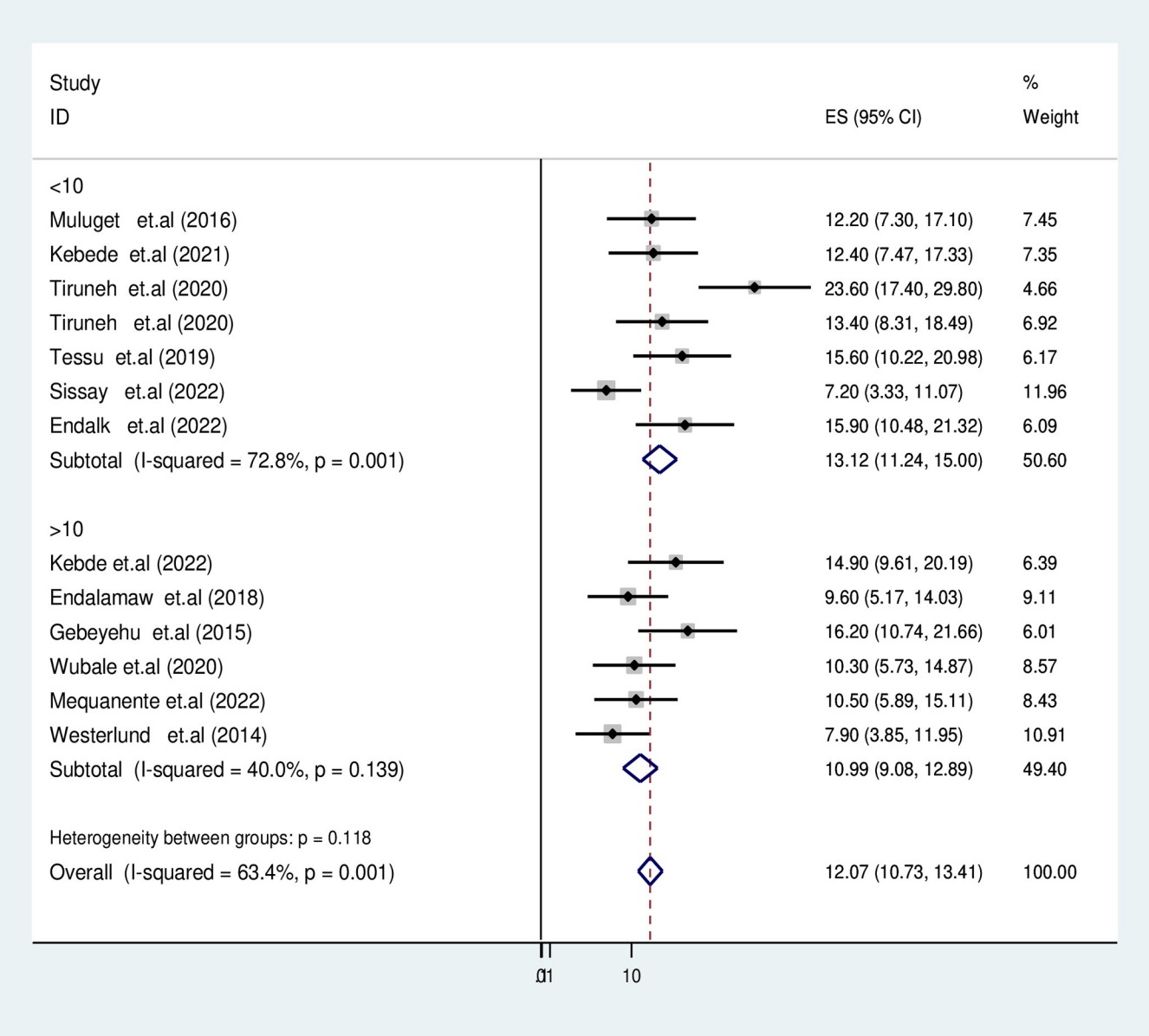

**Fig 4. Forest plot for subgroup analysis by follow-up time of active TB in HIV-infected children in Ethiopia.**

## Publication bias assessment

The publication bias was assessed graphically using funnel plots, and the findings revealed no systematic deviation as depicted in (**Fig 7**).

In addition, Quantitative analysis we had conducted and assessed using Begg's and Egger's tests for biases. Egger's regression was performed, and the report indicated the absence of publication bias for using two factors sample size and follow up periods as elaborated in (**Table 3**).

## Discussion

This systematic review and meta-analysis revealed the pooled burden of active (TB) among HIV-infected children in Ethiopia and further identified predictors associated with active TB.

**Table 2. Factors associated with active TB occurrence among HIV infected children in Ethiopia.**

| Variables | OR | 95%CI | I² | Q² | P value of Q | P value of estimation |
|---|---|---|---|---|---|---|
| Advanced WHO Clinical stages | Ref | Ref | Ref | | Ref | < 0.001 |
| Stages I&II | | | | | | |
| Stage III&IV | 2.27 | [1.18–2.73] | 89.5% | 0.99 | 0.001 | |
| Cotrimoxazole preventive therapy (CPT) | Ref | Ref | Ref | | | < 0.001 |
| Given | | | | | | |
| Not given | 4.26 | [3.47–5.28] | 43.3% | 2.99 | 0.116 | |
| Isoniazid preventive therapy status (IPT) | Ref | Ref | Ref | | | < 0.001 |
| Given | | | | | | |
| Not Given | 2.28 | [1.99–3.18] | 38.8% | 2.07 | 0.22 | |
| Level of Hemoglobin Hgb ≤ 10 gm/dl | 3.11 | (1.57–4.7 | 77.6% | 3.11 | | < 0.001 |
| Hgb > 10 gm/dl | | Ref | | | | |

Hgb = hemoglobin, IPT = Isoniazid preventive therapy status (IPT), Cotrimoxazole preventive therapy (CPT)

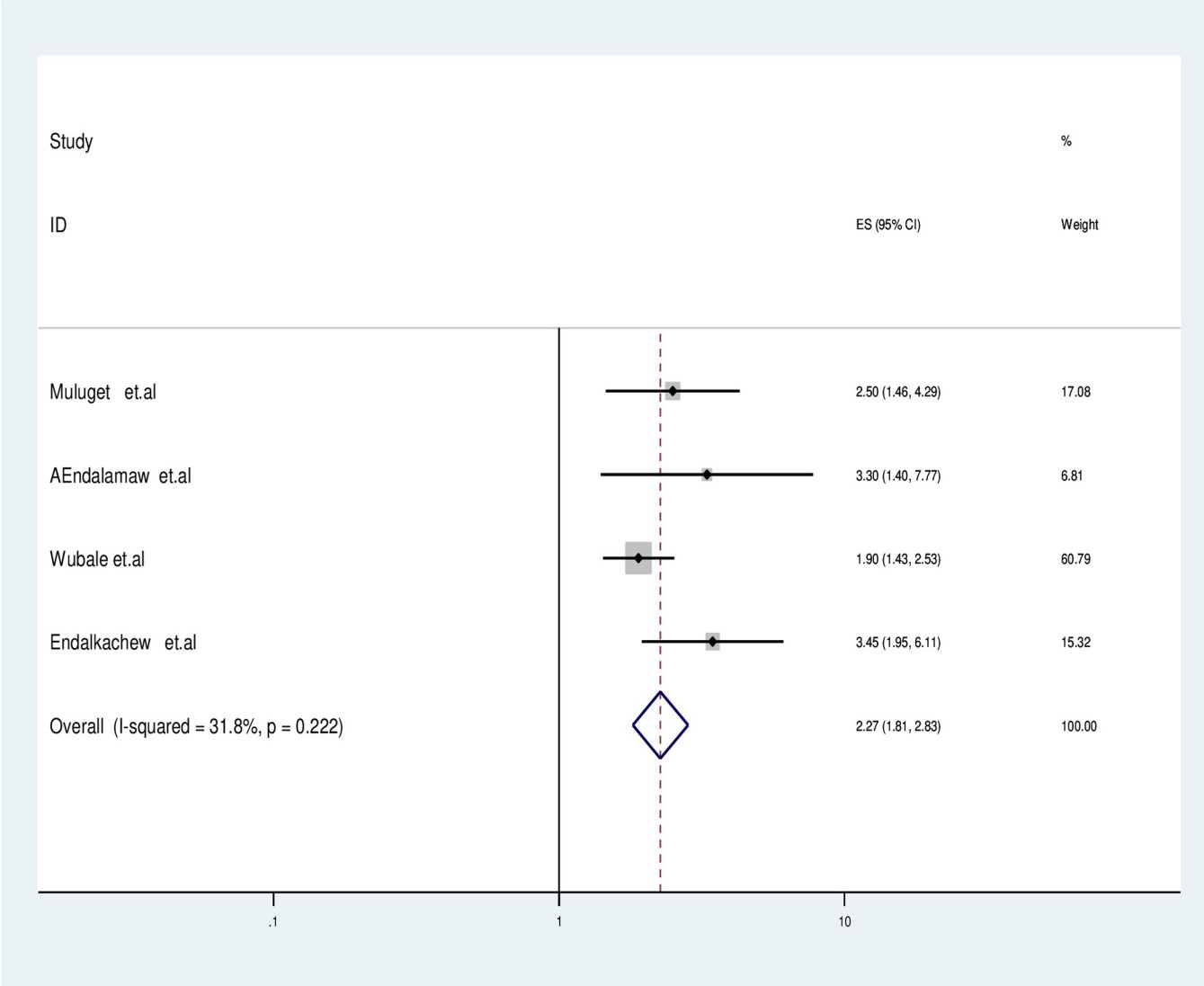

**Fig 5. Forest plotted for impact of missed IPT among active TB in HIV-infected children in Ethiopia.**

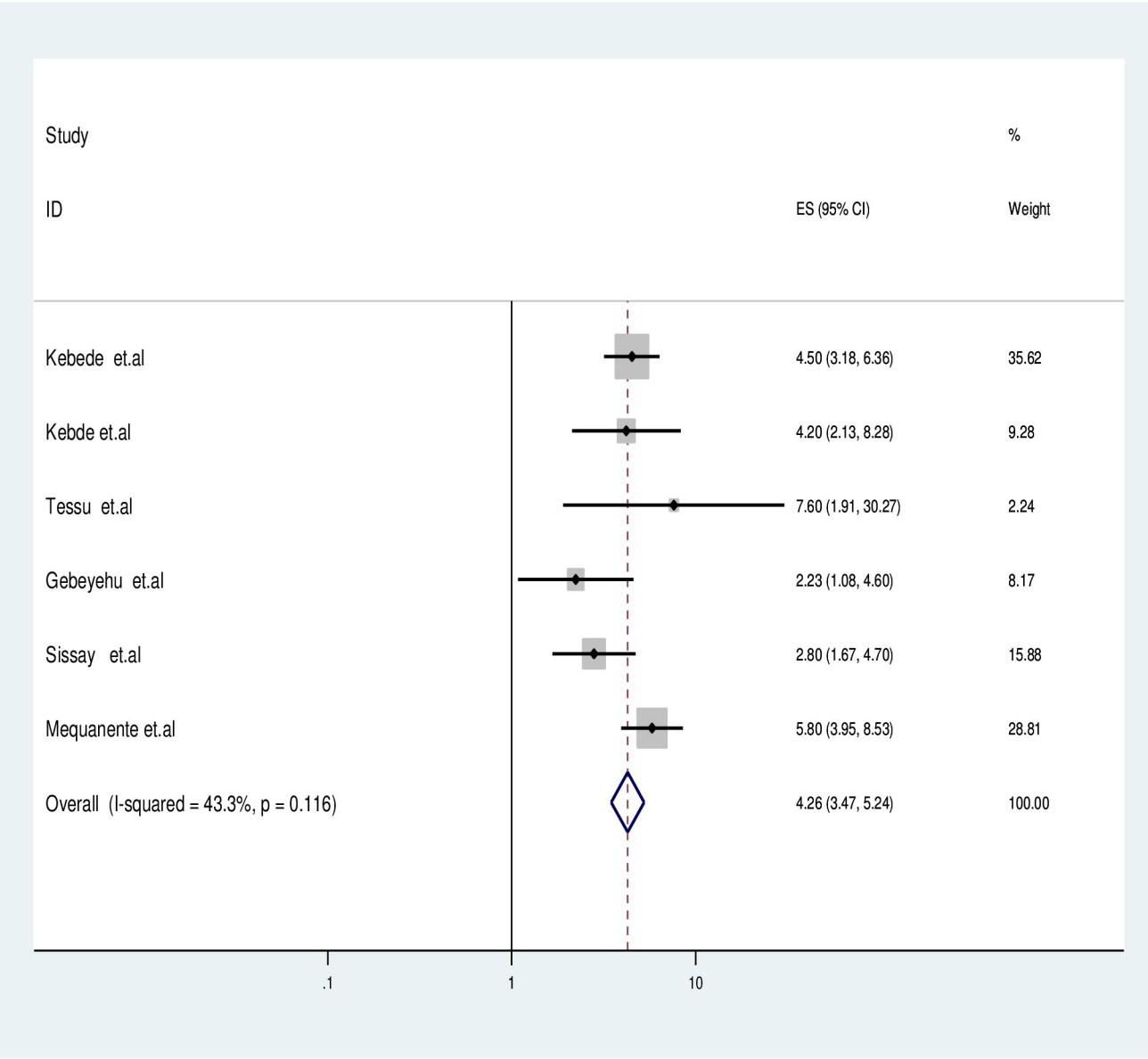

.

**Fig 6. Forest plotted for impact of missed CPT among active TB in HIV-infected children.**

In the final report of 13 individual studies with including 5834 participants, 834 TB and HIV co-infected cases were found at national level. This made the pooled estimated prevalence of active TB was 12.07% (95%CI: 10.73–13.4). This finding is higher than previously reported 0.78% in Ethiopia [40], 43% in SSA countries [41], and 1.03% in Portugal [42]. The findings indicate a significantly high burden of active and need for immediate attention to meet the targets set by the End TB Strategy to achieved the goal of a 90% reduction should to be ($\leq$ 10 TB cases per 100,000 population) by 3035 requires urgent action [1, 2]. Conversely, this report is lower than the previous meta-analysis finding 15% in middle-income countries [43, 44] and

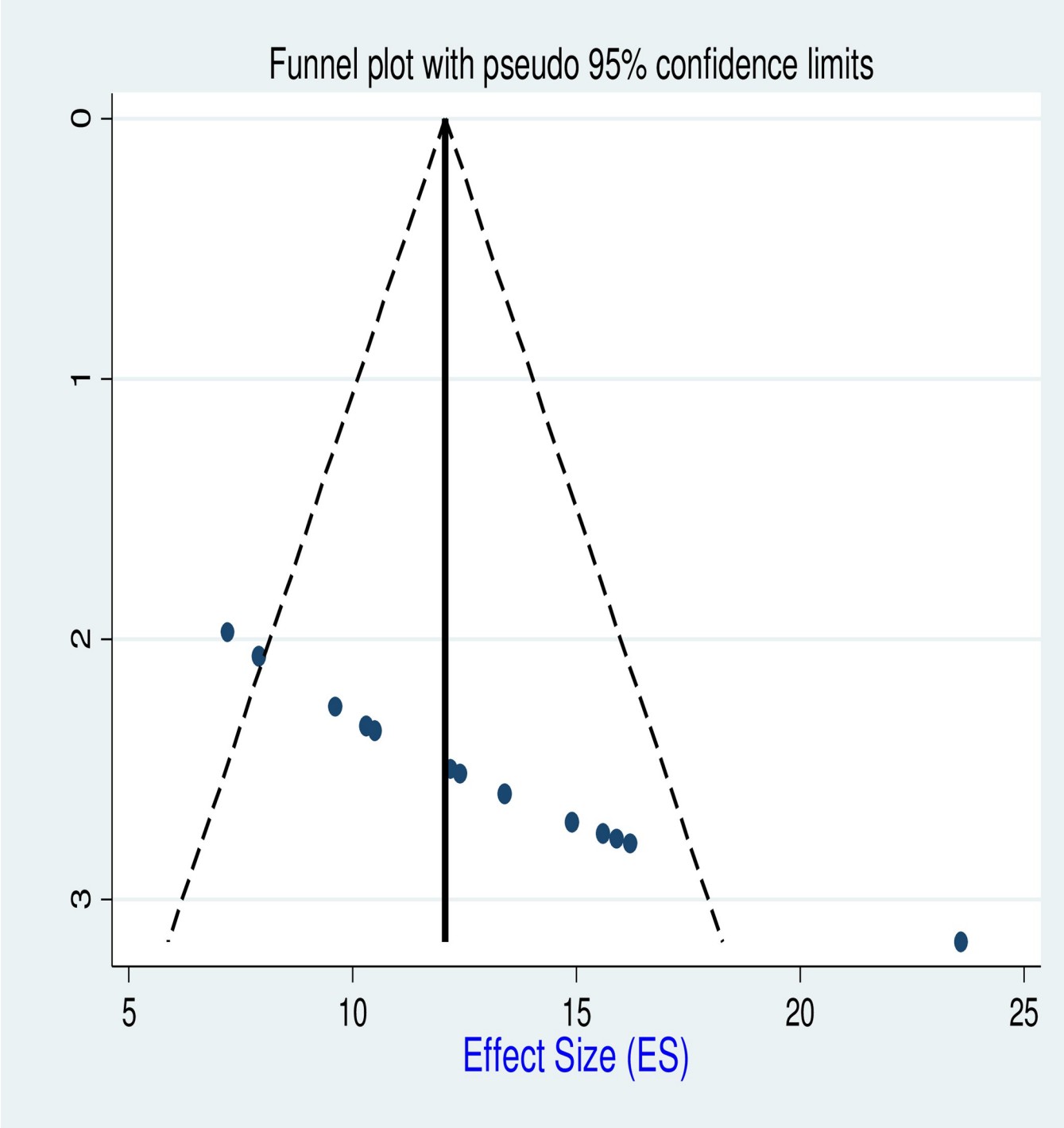

**Fig 7. Funnel plot for active TB proportion among HIV-infected children in Ethiopia.**

32% meta-analysis reported in Nigeria [45]. The variation in the pooled estimate of active TB among the included studies may be attributed to differences in study time TB control measures, and variations among the studies in Nigeria and in Ethiopia [46].Also the variation in healthcare infrastructure, treatment practices, and regional differences also influence pooled prevalence rates.

**Table 3. Potential sources of heterogeneity for pooled active TB burden among HIV infected children in Ethiopia.**

| Logrr | Coefficient | SE | t | p > (t) | P>95%CI |
|---|---|---|---|---|---|
| Sample size | -0.317 | 0.181 | -0.37 | 0.14 | −0.01, 0.015 |
| Follow up periods | −0.443 | 0.512 | - 0.68 | 0.42 | −1.59, 1.089 |
| Constant | 905.22 | 711.3 | 0.311 | 0.15 | −1863.5, 1112.4 |

CI = confidence interval, SE = Standard error

This systematic review and meta-analysis revealed that HIV-infected children with advanced WHO clinical stage (III&IV) have a twofold increased likelihood of experiencing active TB compared with mild advanced WHO clinical stage. This finding is supported by previous meta-analyses conducted in Ethiopia [47–50]. This possible justification for this finding is that children with advanced HIV disease may have compromised immune systems due to their clinical stage III&IV, which is associated with low CD4 counts and could lead to an increased risk of developing opportunistic infections, including TB [51].

The report of this meta-analysis revealed that, the risk of developing active TB was four fold increased for HIV infected children with Hgb ≤10 gm/dl than in those with Hgb >10 gm /dl. This is consistent with the previous study finding [52–54]. This could be attributed to the fact that anemia can indeed impair the immune response and the bactericidal activity of leukocytes, making individuals more vulnerable to infections, including tuberculosis.

The odds of developing active TB among HIV infected children who missed CPT had a four-fold risk as compared ever given children. This is consistent with previously reported meta-analysis in finding in Ethiopia [53, 55, 56]. This might be due to cotrimoxazole, is prescribed to HIV-infected children to prevent lethal opportunistic by preventing production of nucleic acids and proteins essential for the growth of opportunistic infections including PCP, and toxoplasmosis thus helping to counteract immunosuppression and disease progression.

Consistent with previous studies finding in [12, 17, 32, 33, 38, 57–60],concurrent administering of IPT after ruled-out of active TB symptoms with ART demoted more than 90% of active TB-associated incidence of morbidity [3, 61, 62]. In the final report of this systematic review and meta-analysis, it was found that HIV-infected children who did not receive IPT (preventive therapy) were at a twofold higher risk compared to the control group. This might be IPT (Isoniazid Preventive Therapy) has the potential to reduce the burden of latent mycobacteria in the lungs. This is because Isoniazid preventive therapy can effectively stop the progression of latent TB infection from developing into active TB disease [17, 63].

In contrast to previous systematic review findings [53, 55] and primary studies reported [12, 17, 32, 33, 38, 57–60] this meta-regression found no significant association between declined CD4 count (≤200 cells/cml), age of patients, duration of follow-up, comorbidity status, and functional status with the risk developing active TB in HIV co-infection children. This might be related to the methodological differences, heterogeneity of included study populations, sample size limitations, publication bias, unaccounted factors, and further experimental studies are highly needed to better understand this relationship.

## Strengths and limitations of the study

The strengths of this study include an extensive search strategy, clear inclusion criteria, and the involvement of five independent authors in the quality, inclusion and analysis for this systematic review and meta-analysis. However, there are several methodological limitations including focusing on articles published only on English were included and the extracted articles were from four Ethiopian regions were included in the analysis, such that some of the

region may not be represented. In addition, limitations such as reliance on clinical stratification or non-laboratory-supported staging, sub-standard diagnostic capacities in health facilities, a small number of included studies, and the use of retrospective data may potentially affect the validity of the results.

## Conclusion and recommendation

This systematic review found a higher prevalence of active TB in HIV-infected children in Ethiopia compared to the estimated rates in the end TB strategy. Risk factors for active TB were identified included WHO clinical stages IV and III, low hemoglobin, missed IPT, and missed CPT were predictors. To reduce the risk of active TB, it is crucial to implement effective strategies such as regular IPT mission and addressing the gaps in treatment, and routine screening for active TB during follow-ups to prevent premature death.

## Supporting information

**S1 Checklist. PRISMA 2020 chiecklist.**
(DOC)

**S1 Text. Article searching strategy for one of PubMed date base.**
(DOCX)

**S1 Table. The JBI quality assessment check list for included studies.**
(DOCX)

**S2 Table. Minimal data set for this met-analysis.**
(XLSX)

## Acknowledgments

### Author contributions

**Fassikaw Kebede Bizune's**; Conceptualization, Data curation, Formal analysis, Funding acquisition, Investigation, Methodology, Project administration, Resources, Software, Supervision, Validation, Visualization, Writing–original draft, Writing–review & editing.

**Dejen Tsegaye's**; Conceptualization, Data curation, Formal analysis, Funding acquisition, Investigation, Methodology, Project administration, Resources, Software, Supervision, Validation, Visualization.

**Belete Negese;** Resources, Software, Supervision, Validation, Visualization, Writing -review & editing.

**Tsehay Kebede's Bizueneh:** Conceptualization, Data curation, Formal analysis, Funding acquisition, Investigation, Validation, Visualization, Writing–original draft, Writing–review & editing.

## Author Contributions

**Conceptualization:** Fassikaw Kebede Bizuneh, Dejen Tsegaye, Tsehay Kebede Bizuneh.

**Data curation:** Fassikaw Kebede Bizuneh, Dejen Tsegaye, Tsehay Kebede Bizuneh.

**Formal analysis:** Fassikaw Kebede Bizuneh, Dejen Tsegaye, Tsehay Kebede Bizuneh.

**Funding acquisition:** Fassikaw Kebede Bizuneh, Dejen Tsegaye, Tsehay Kebede Bizuneh.

**Investigation:** Fassikaw Kebede Bizuneh, Dejen Tsegaye, Tsehay Kebede Bizuneh.

**Methodology:** Fassikaw Kebede Bizuneh, Dejen Tsegaye.

**Project administration:** Fassikaw Kebede Bizuneh, Dejen Tsegaye.

**Resources:** Fassikaw Kebede Bizuneh, Dejen Tsegaye, Belete Negese Gemeda.

**Software:** Fassikaw Kebede Bizuneh, Dejen Tsegaye, Belete Negese Gemeda.

**Supervision:** Fassikaw Kebede Bizuneh, Dejen Tsegaye, Belete Negese Gemeda.

**Validation:** Fassikaw Kebede Bizuneh, Dejen Tsegaye, Belete Negese Gemeda, Tsehay Kebede Bizuneh.

**Visualization:** Fassikaw Kebede Bizuneh, Dejen Tsegaye, Belete Negese Gemeda, Tsehay Kebede Bizuneh.

**Writing – original draft:** Fassikaw Kebede Bizuneh, Tsehay Kebede Bizuneh.

**Writing – review & editing:** Fassikaw Kebede Bizuneh, Belete Negese Gemeda, Tsehay Kebede Bizuneh.

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
