## [Decision Letter · Decision Letter 0]

4 Feb 2024

PGPH-D-23-02361

ACTIVE TUBERCULOSIS PREVALENCE AFTER ANTI-RETROVIRAL THERAPY AMONG SEROPOSITIVE CHILDREN LIVING IN ETHIOPIA: A SYSTEMATIC REVIEW AND META-ANALYSIS

Dear Dr. Fassikaw Kebede

Thank you for submitting your manuscript to PLOS Global Public Health. After careful consideration, we feel that it has merit but does not fully meet PLOS Global Public Health’s publication criteria as it currently stands. Therefore, we invite you to submit a revised version of the manuscript that addresses the points raised during the review process.

We look forward to receiving your revised manuscript.

Kind regards,

Leonardo Martinez

Academic Editor

Journal Requirements:

1. We noticed you have some minor occurrence of overlapping text with the following previous publication(s), which needs to be addressed:

- https://doi.org/10.1155/2022/9925693

In your revision ensure you cite all your sources (including your own works), and quote or rephrase any duplicated text outside the methods section. Further consideration is dependent on these concerns being addressed.

2. We noticed that you used "unpublished" in the manuscript. We do not allow these references, as the PLOS data access policy requires that all data be either published with the manuscript or made available in a publicly accessible database. Please amend the supplementary material to include the referenced data or remove the references.

3. Please provide separate figure files in .tif or .eps format only and remove any figures embedded in your manuscript file. Please also ensure all files are under our size limit of 10MB.

Reviewers' comments:

Reviewer's Responses to Questions

**Comments to the Author**

1. Does this manuscript meet PLOS Global Public Health’s publication criteria? Is the manuscript technically sound, and do the data support the conclusions? The manuscript must describe methodologically and ethically rigorous research with conclusions that are appropriately drawn based on the data presented.

Reviewer #1: Partly

Reviewer #2: Yes

Reviewer #3: Partly

2. Has the statistical analysis been performed appropriately and rigorously?

Reviewer #1: No

Reviewer #2: No

Reviewer #3: Yes

3. Have the authors made all data underlying the findings in their manuscript fully available (please refer to the Data Availability Statement at the start of the manuscript PDF file)?

Reviewer #1: No

Reviewer #2: Yes

Reviewer #3: No

4. Is the manuscript presented in an intelligible fashion and written in standard English?

Reviewer #1: Yes

Reviewer #2: Yes

Reviewer #3: No

5. Review Comments to the Author

Reviewer #1: All comments given in attachment.

Q1 and Q2 - main concerns - there may be one duplicate study population - researchers need to verify that this is not the case.

Incidence data analysis incomplete. The rest of the analyses is done fairly comprehensively and appropriately.

Q3. I could not find the corresponding data file in supplementary files - although the pdf manuscript states that the data was provided.

Full reviewer comment report is provided as an attachment.

Reviewer #2: Abstract

Since you have a relatively well articulated burden of the problem in the introduction better to reflect some summary figures in the background of the abstract for better conveyance of your studies relevance

Introduction

you have tried to discuss on the burden of the problem and its impact in a clear manner, i perceive the introduction lacks highlighting global and national policies and initiatives undertaken to address the problem. and hopefully this will set up a ground for your studies significance and engages readers more on the impact of your studies

Data synthesis and analysis procedure

If i am not mistaken, p value <0.1 actually represents significant heterogeneity whether for overall hetrogenity assessment of sub group analysis we use Q test and

Ho - No difference between studies

H1 - Difference between studies or groups

And we reject the null hypothesis when p value is significant (,0.05 or preferably 0.1 since the test has

low power. if you used opposite cut offs i believe it affects your studies significantly, but if its just a typo you could just correct here

Result(Associated factors)

Regarding the associated factors;

1- You need to reconsider the thematic clustering you have used or at least report the specific factors associated in each study and how you have clustered each, to ensure transparency

2. Since Active Tb is one of the staging criteria to stage a patient on WHO clinical staging schema, i have strong reservation regarding using WHO stage as an associated factor since by implication it contains the outcome variable , hence creating a kind of falsely inflated Beta coefficient.

3. You mentioned that WHO stage and IPT were statistically significant but you resorted to report OR for anemia and stage dropping IPT. you also dropped IPT on further discussions so please review this since it has critical implication on you other results as well.

Discussion

I think you should carefully review your discussion to include strong discussion points about reason you impute for your results, possible implications of the result especially on preventive therapies for seropositive children and furthure interventions you recommend the criticality of addressing anemia in this group of patients. Moreover, you might also go indepth on similar feature of the studies which might affect the results ...for ex you mentioned diagnostic methods of the studies ....you could further elaborate on that and compare it to global and national recommendations and the possible implication your result might have on diagnostic methods ( using more sensitive

diagnostic methods for identified at risk groups and so on)

Strength and Limitation

Given the challenges of Tb diagnosis specially in resource limited set ups like ours i expect you to delve more in to the scenarios surrounding the studies you pooled which might hamper the validity of your results. one such factor would be mostly clinical stratification of seropostive patients or nonlab-supported staging, mostly, and sub-standard

diagnostic capacities of our health facilities

Reviewer #3: At the outset, I must congratulate the authors on their work in the field of HIV and Tuberculosis, both very serious debilitating infections which have significant impact on each other as co-infections.

I must highlight the strengths of the study :

a) Well chosen topic for meta-analysis and SR with large sample inclusion criteria hence improving your sample size for adequate representation.

b) Very methodic plan of action followed for the review as documented in the study.

c) Strong conclusion with a significant outlook towards engaging more work in the field and more action in the form of change in plans and educating doctors and health care professionals involved in the care of children living with HIV.

Limitations and Suggestions:

a) Use standard terminologies across the entire study and please abbreviate with explanations the first time.

b) The methodology is repeated about 4 times in the paper and hence becomes repetitive and cumbersome and hence would request you to kindly review and see how it may be framed better.

c) Multiple spelling and grammatical errors to be addressed.

d) References have not been done as per the required formatting and hence the data used for study purpose has not been reviewed by me.

e) Multiple factual errors and statements with no scientific backing have been mentioned, there appears to be gross errors indicative of possible cut and copy approach from a previously done study.

f) References and data regarding prevalence of HIV, TB etc are all outdated and hence need to be updated accordingly.

I have attached a word document with tracked changes and comments for which I am hoping to get a reply prior to final submission. I am confident of your expertise to be able to re work this paper and make it fit to be published for maximal impact.

6. PLOS authors have the option to publish the peer review history of their article (what does this mean?). If published, this will include your full peer review and any attached files.

**Do you want your identity to be public for this peer review?** For information about this choice, including consent withdrawal, please see our Privacy Policy.

Reviewer #1: No

Reviewer #2: **Yes: **Henok Tadesse Bireda

Reviewer #3: **Yes: **Dr. Nikith D'Souza MD DNB

---

## [Decision Letter · Decision Letter 1]

10 May 2024

PGPH-D-23-02361R1

Meta-Analysis of Active Tuberculosis Occurrence among Children Living with HIV Post Anti-Retroviral Therapy Initiated in Ethiopia

Dear Dr. Kebede,

Thank you for submitting your manuscript to PLOS Global Public Health. After careful consideration, we feel that it has merit but does not fully meet PLOS Global Public Health’s publication criteria as it currently stands. Therefore, we invite you to submit a revised version of the manuscript that addresses the points raised during the review process.

Please note that the reviewers have provided attachments with suggestions and feedback for you to consider and address.

We look forward to receiving your revised manuscript.

Kind regards,

Sanghyuk S Shin

Academic Editor

Journal Requirements:

1. We noticed that you used "unpublished" in the manuscript. We do not allow these references, as the PLOS data access policy requires that all data be either published with the manuscript or made available in a publicly accessible database. Please amend the supplementary material to include the referenced data or remove the references.

Additional Editor Comments (if provided):

Reviewers' comments:

Reviewer's Responses to Questions

**Comments to the Author**

1. If the authors have adequately addressed your comments raised in a previous round of review and you feel that this manuscript is now acceptable for publication, you may indicate that here to bypass the “Comments to the Author” section, enter your conflict of interest statement in the “Confidential to Editor” section, and submit your "Accept" recommendation.

Reviewer #1: (No Response)

Reviewer #3: All comments have been addressed

2. Does this manuscript meet PLOS Global Public Health’s publication criteria? Is the manuscript technically sound, and do the data support the conclusions? The manuscript must describe methodologically and ethically rigorous research with conclusions that are appropriately drawn based on the data presented.

Reviewer #1: No

Reviewer #3: Partly

3. Has the statistical analysis been performed appropriately and rigorously?

Reviewer #1: Yes

Reviewer #3: Yes

4. Have the authors made all data underlying the findings in their manuscript fully available (please refer to the Data Availability Statement at the start of the manuscript PDF file)?

Reviewer #1: Yes

Reviewer #3: Yes

5. Is the manuscript presented in an intelligible fashion and written in standard English?

Reviewer #1: Yes

Reviewer #3: No

6. Review Comments to the Author

Reviewer #1: DETAILED REVIEWER COMMENTS WERE PROVIDED AS AN ATTACHMENT ON JANUARY 19TH 2024 (36 COMMENTS).

THE AUTHORS SEEM TO HAVE FAILED TO SEE THE ATTACHMENT, THEY HAVE NOT RESPONDED TO THE 36 COMMENTS.

THEY RESPONDED TO GENERAL COMMENTS OF THE REVIEWER.

THE ATTACHMENT HAS BEEN UPLOADED AGAIN FOR THE AUTHORS ATTENTION.

IF THESE MAJOR ISSUES ARE ADEQUATELY ADDRESSED THE PAPER WILL BE SUITABLE FOR PUBLICATION.

Reviewer #3: Thank you for addressing the comments and re-working the paper to match the requirements of the journal. Here are few comments which help bring out the true impact of your research:

Must appreciate:

a) Updating the paper with 2022 and 2023 articles and .

b) Better structuring of the methodology.

c) Results are better highlighted.

d) Stronger conclusion and recommendations.

Could work on the following:

a) Review the spelling errors as indicated in the attachment in the previous and current review.

eg Whorl Health Organisation for World Health Organisation.

Human Immune Deficiency Varies for Human Immunodeficiency Virus

Forest Plotted for Forest Plot

b) Gross errors like use of Hgb <10mg/dl instead of 10gm/dl in the paper are serious oversights by the team of authors! Please review and rectify.

c) Results to be provided as absolute facts based on your findings, avoid use of terminologies like "more than half", "most" and "Almost all".

d) Conclusions require adequate referencing to prevent it to seem like arbitrary statements and opinions.

e) Title and Short title require work to be done - Title speaks of incidence of TB, however in your methodology there is use of prevalence. Title is not clear and concise and short title is incomplete.

f) Few of the results and inferences like CPT do not have p values provided for the AOR;

g) Rephrasing of the results and inferences. There is repetition between CPT and IPT.

7. PLOS authors have the option to publish the peer review history of their article (what does this mean?). If published, this will include your full peer review and any attached files.

**Do you want your identity to be public for this peer review?** For information about this choice, including consent withdrawal, please see our Privacy Policy.

Reviewer #1: No

Reviewer #3: **Yes: **Nikith Austin DSouza MD DNB

---

## [Decision Letter · Decision Letter 2]

5 Jul 2024

Active Tuberculosis Prevalence in HIV-Infected Children on Antiretroviral Therapy in Ethiopia: A Systematic Review

PGPH-D-23-02361R2

Dear Mr Kebede,

We are pleased to inform you that your manuscript 'Active Tuberculosis Prevalence in HIV-Infected Children on Antiretroviral Therapy in Ethiopia: A Systematic Review' has been provisionally accepted for publication in PLOS Global Public Health.

Best regards,

Sanghyuk S Shin

Academic Editor

Reviewer Comments (if any, and for reference):

Reviewer's Responses to Questions

**Comments to the Author**

1. If the authors have adequately addressed your comments raised in a previous round of review and you feel that this manuscript is now acceptable for publication, you may indicate that here to bypass the “Comments to the Author” section, enter your conflict of interest statement in the “Confidential to Editor” section, and submit your "Accept" recommendation.

Reviewer #1: All comments have been addressed

2. Does this manuscript meet PLOS Global Public Health’s publication criteria? Is the manuscript technically sound, and do the data support the conclusions? The manuscript must describe methodologically and ethically rigorous research with conclusions that are appropriately drawn based on the data presented.

Reviewer #1: Yes

3. Has the statistical analysis been performed appropriately and rigorously?

Reviewer #1: Yes

4. Have the authors made all data underlying the findings in their manuscript fully available (please refer to the Data Availability Statement at the start of the manuscript PDF file)?

Reviewer #1: Yes

5. Is the manuscript presented in an intelligible fashion and written in standard English?

Reviewer #1: Yes

6. Review Comments to the Author

Reviewer #1: I congratulate the authors for carefully considering all the comments and suggestions, they have adequately addressed the issues raised in this revised manuscript. This manuscript provides a robust and well analysed systematic review and meta-analysis on TB in CLHIV in Ethiopia. This provides valuable evidence on the continued high burden of TB in this vulnerable population in an African setting.

7. PLOS authors have the option to publish the peer review history of their article (what does this mean?). If published, this will include your full peer review and any attached files.

**Do you want your identity to be public for this peer review?** For information about this choice, including consent withdrawal, please see our Privacy Policy.

Reviewer #1: **Yes: **Elizabeth Maleche-Obimbo
